# Impact of Universal Test and Treat (UTT) on anticipated stigma among patients newly diagnosed with HIV in Johannesburg, South Africa: A cross-sectional study

**Tembeka Sineke**[1,2], **Idah Mokhele**[1,2], **Robert A.C. Ruiter**[2], **Mandisa Dukashe**[3,4], **Dorina Onoya**[1*]

**1** Health Economics and Epidemiology Research Office, Department of Internal Medicine, School of Clinical Medicine, Faculty of Health Sciences, University of the Witwatersrand, Johannesburg, South Africa, **2** Department of Work and Social Psychology, Maastricht University, Maastricht, The Netherlands, **3** The South African National Aids Council, National Strategic Plan Unit, Hatfield, South Africa, **4** HIV Survivors and Partners Network, Centurion, South Africa

* donoya@heroza.org

## Abstract

Anticipated stigma—the fear that an HIV diagnosis and disclosure may lead to negative social consequences that undermine engagement in HIV care, even in the era of universal test-and-treat (UTT). This study aimed to estimate the prevalence of anticipated stigma and identify its predictors among newly HIV-diagnosed adults in Johannesburg, South Africa, where treatment is available. We conducted a cross-sectional survey among 652 adults (≥18 years) newly diagnosed with HIV between October 2017 and August 2018 at four primary healthcare clinics in Johannesburg. Participants (64.1% female; median age 33 years, IQR 28–39) were interviewed immediately after receiving their HIV test results. Anticipated stigma was measured using an adapted five-item, four-point scale assessing concerns related to HIV disclosure and concealment (Cronbach's alpha = 0.82). Mean scores were categorized as low-to-medium (≤2.5) or high (>2.5). Modified Poisson regression was used to identify predictors of high anticipated stigma, reported as adjusted risk ratios (aRRs) with 95% confidence intervals (CIs). Overall, 55% of participants reported high anticipated stigma. Prevalence was higher among males (55.8%) and young adults aged 18–29 years (61.1%), and lower among married individuals (43%). Compared to married participants, those in unmarried relationships were more likely to report high anticipated stigma (aRR 1.10, 95% CI: 1.01–1.18). Lower anticipated stigma was observed among older individuals (aRR 0.94 for ages 30–39 vs 18–29), those whose primary home was in another province or country, those living in their current home for five or more years, individuals with fewer ART-related concerns, and those reporting lower perceived social support. Despite universal access to HIV treatment, over half of newly diagnosed adults reported high anticipated stigma.

**Data availability statement:** The de-identified data supporting the findings of this study have been uploaded data and code in a public repository (https://figshare.com/articles/dataset/Impact_of_Universal_Test_and_Treat_UTT_on_expected_stigma_among_patients_newly_diagnosed_with_HIV_in_Johannesburg_South_Africa_a_cross-sectional_study/30157579).

**Funding:** This study was made possible by the generous support of the American People and the President's Emergency Plan for AIDS Relief (PEPFAR) through US Agency for International Development (USAID) under the terms of Cooperative Agreements AID-674-A-12-00029 and 72067419CA00004 to Health Economics and Epidemiology Research Office and under the terms of Cooperative Agreement 674-A-00-09-00018-00 to Boston University. We were also supported by the grant 1R01AI152149-01 from the National Institute of Health (NIH). The contents are the responsibility of the authors and do not necessarily reflect the views of the NIH, PEPFAR, USAID or the United States Government.

**Competing interests:** The authors have declared that no competing interests exist.

These findings underscore the need for targeted interventions to address persistent drivers of stigma, strengthen coping skills, and support social integration to promote engagement in HIV care.

## Introduction

South Africa has made substantial progress towards achieving the 95-95-95 targets set by UNAIDS [1], reaching 95.4% of people living with HIV (PLHIV) diagnosed, of those diagnosed, 78.7% on Antiretroviral Treatment (ART), and 91.3% of those on treatment achieving viral suppression as of 2023 [2]. In recent years, access to ART is widespread, and HIV has become more of a chronic health condition as people living with HIV (PLHIV) can now live longer. However, while same-day ART initiation has been an essential step in increasing accessibility [3,4], it does not directly address social challenges that PLHIV continue to face [5]. While treatment improves the personal health and longevity outlook of the individual living with HIV, it does not address the social reintegration needed to facilitate the return to normal social interactions.

HIV denialism previously fuelled stigma, questioning the link between HIV and AIDS, and delaying the rollout of ART [6]. This was coupled with misinformation, shame, and reluctance to disclose one's status [7]. Internalised stigma, characterised by feelings of low self-worth, guilt, and shame, persists [8][9]. It often results in fear of disclosure, self-blame, and recent reports have revealed an overall pooled prevalence to be 35.7% among people living with HIV in an African setting [10]. This internal stigma fuels anticipated social stigma, resulting in self-isolation that has a detrimental impact on PLHIV's quality of life. Recent work has shown that PLHIV still anticipate and experience stigma in the context of romantic relationships (starting or maintaining) [11].

While ART access has improved in South Africa, treatment disengagement and inconsistent adherence to ART remain a threat to achieving sustained viral suppression and eliminating HIV in resource-limited settings [12–14]. Anticipated stigma is defined as a belief and expectation by PLHIV of future repercussions and ill treatment due to their HIV positive status. PLHIV who anticipate stigma may withdraw from social relationships in an attempt to minimise potential discrimination, leading to social isolation and withdrawal from potentially supportive networks [15]. Among other factors, anticipated stigma is a significant predictor of poor adherence to ART [16–18]. Anticipated stigma has also been associated with distrust of healthcare workers [19], increased medication concerns, and low treatment knowledge. PLHIV also fear that engaging in ART care may unintentionally reveal one's HIV status through regular clinic visits and carrying medication or printed medical information [15]. PLHIV have voiced concerns that unintentional disclosure would result in stigmatisation [20].

Other studies have shown that anticipated stigma negatively impacts testing for HIV, a barrier to ART initiation, and several studies have shown HIV-related stigma

to be associated with depressive symptoms among PLHIV [21,22]. This may result in social withdrawal and isolation as individuals tend to suppress their emotions due to fears of being stigmatised [23].

The National Strategic Plan (NSP) for 2023 identified stigma as one of the important issues to be addressed. It called for interventions to address this [24]. These include evidence-based interventions, community engagement, and collaborations between different sectors. In light of the changes in treatment guidelines which include the country progressively raising the CD4-based ART eligibility threshold from 500 cells/μL (2015) and September 2016, it removed the CD4 threshold entirely and adopted the WHO Universal Test and Treat (UTT) policy, making all people living with HIV eligible for ART immediately upon diagnosis [3,4], we aimed to determine the prevalence and predictors of anticipated stigma among newly HIV-diagnosed individuals under the UTT policy in Johannesburg, South Africa.

Understanding the prevalence and predictors of anticipated stigma in this context is therefore crucial for informing targeted interventions, supporting treatment adherence, and achieving national HIV targets. A study in Umlazi Township found that 30.8% of participants reported anticipated HIV stigma, with higher prevalence among women and individuals living with HIV [25]. This study aims to provide data on anticipated stigma in Johannesburg, thus contributing to the evidence base needed to inform stigma-reduction strategies and improve HIV care outcomes in line with the NSP's objectives

## Materials and methods

### Ethics statement

The study protocol was reviewed and approved by the Human Research Ethics Committee of the University of Witwatersrand (M1704122). Written informed consent was obtained from all participants prior to their inclusion in the study. All personal identifiers were removed from the final analytic dataset.

### Study population and procedures

The study enrolled newly diagnosed adult participants (≥18 years) from October 2017 to August 2018 at four primary healthcare clinics (PHCs) in Johannesburg, South Africa. The enrolment of patients was conducted according to previously described criteria [26]. Eligible participants who provided written informed consent prior to study participation completed a structured baseline questionnaire on the day of HIV diagnosis. Of the 708 patients who tested positive during the study period,703 (99.3%) newly diagnosed patients were successfully referred and screened, 652 (92.7%) were eligible and agreed to participate in the study. Patients were interviewed immediately after HIV diagnosis, and ART initiation was determined through medical record review up to six months post-test. While the parent study was designed as a longitudinal cohort, the present analysis focuses specifically on the baseline cross-sectional data examining anticipated HIV stigma among newly diagnosed adults. Participants self-reported being newly diagnosed during the screening process. Enrolled patients with a prior history of ART were excluded from the analytic dataset. We also excluded patients who were psychologically unable or too sick to participate, unwilling to provide consent or planned to get treatment elsewhere. Additionally, women who were pregnant at HIV diagnosis were excluded from the study because in-pregnancy treatment initiation and care processes differ from those of non-pregnant women.

### Variable definitions

We used an adapted five-item, four-point scale (1-"strongly agree" to 4-"strongly disagree") measuring agreement with statements regarding HIV disclosure concerns and HIV status concealment. Questions included concerns regarding rejection, fear of being judged, concerns about the risk of disclosure to others, feelings of shame regarding the diagnosis, and a need to keep the diagnosis as a secret (S1 Text). All questions were closed-ended with four-point response options. For the final analysis, we dichotomized the final outcome of anticipated stigma whereby mean scores were categorized as "low-to-medium" (if the score<=2.5), or "high" (if the score>2.5) [27,8] defining anticipated

stigma with a four-point scale measuring agreement with statements regarding HIV disclosure concerns and HIV status concealment (Cronbach's alpha = 0.82). Depression was measured using the CES-D 10 scale, a 10-question four-point scale (which had scores ranging from 0 to 3) that measures general depressive symptoms experienced up to seven days before the interview [28]. The total score ranged from zero to 30, with higher scores reflecting higher depression (Cronbach's alpha = 0.81). Depression was dichotomized into major depression (total score of ≥ 12) and no major depression (total score < 12) [29,30].

We created a household amenities index to measure participants' access to 13 key amenities, including basic services (e.g., flush toilet, electricity, gas), housing quality, household density, food availability, and ownership of durable assets (e.g., TV, radio, refrigerator, phone, microwave, computer). The index showed good internal consistency (Cronbach's alpha = 0.79) [31]. Concerns about ART were assessed using a 12-item, 4-point Likert scale questionnaire (Cronbach's alpha = 0.83), covering worries about long-term effects, side effects, and perceived need for ART. Mean scores were categorized as low (<2), medium (2–3), or high (>3) concerns. Perceived social support (PSS) was measured with an 8-item satisfaction scale (Cronbach's alpha = 0.61), and this measured participants' satisfaction with the support made available to them [31], with mean scores categorized as low (<2), medium (2–<3), or high (≥3).

Other socio-demographic factors assessed included age, sex, highest education completed, marital status, employment status, whether the patient was the household breadwinner, the number of child dependents the patient had, and primary source of income. Factors related to health care access, including the history of visiting the testing clinic or any other health provider, and HIV testing history were also assessed. Additional factors also included a history of sexual risk behaviour, including condom usage at last sex, number of sexual partners in the previous twelve months and to whom the patients had disclosed their intention to come for HIV testing, and whether the patients were accompanied by anyone to the testing clinic [26].

## Statistical analysis

We describe the characteristics of study participants using proportions, frequencies, means with standard deviation (SD), and medians with interquartile ranges (IQRs) as appropriate. Factors associated with the prevalence of high anticipated stigma among newly diagnosed patients were assessed using the Modified Poisson regression and report adjusted risk ratios (aRR) with 95% confidence intervals (CIs). The Modified Poisson [32,33] was appropriate for estimating relative risks since the prevalence of anticipated stigma was > 10%. Data analysis was conducted using STATA version 14 (StataCorp, College Station, TX).

## Results

### Demographic and clinical characteristics of newly diagnosed HIV patients

A total of 652 newly HIV diagnosed participants were enrolled (Table 1). Participants had a median age of 33 years (IQR: 28.0-39.0), 64.1% were female, and 35.3% were married. Over half of the participants (51.5%) spoke Nguni languages (which include: Zulu, Xhosa, Ndebele, Swati, etc.). More than half of the participants were employed. Additionally, only 62.4% had disclosed their intention to test for HIV before the clinic visit and 26.0% were accompanied to the testing site. However, participants' baseline intention to start ART was nearly universal at 99.4%, with only 0.6% planning to never start ART. Similarly, 96.3% of participants intended to disclose their HIV-positive status, 4.7% had low perceived social support, and 52% had medium concerns about ART.

### High anticipated stigma among newly diagnosed patients

Overall, 55% of the participants were classified as having high anticipated stigma, while 45% were classified as having low to medium anticipated stigma (Fig 1). Among male participants, 56% had high anticipated stigma compared to 54% of females

**Table 1. Participant sociodemographic characteristics (n = 652).**

|  | Total |
|---|---|
|  | N = 652 (col%) |
| **Age at HIV diagnosis, years Median (IQR)** | 33 (24-39) |
| 18-29.99 | 205 (31.4) |
| 30-39.99 | 293 (44.9) |
| 40+ | 154 (23.6) |
| **Marital status** |  |
| Married | 92 (14.1) |
| In a relationship (living together) | 230 (35.3) |
| In a relationship (not living together) | 202 (31.0) |
| Not in a relationship | 127 (19.5) |
| **Highest education level** |  |
| Primary school or less | 91 (14.0) |
| Some secondary school | 384 (58.9) |
| >=Grade 12 | 177 (27.1) |
| **Employment status** |  |
| Employed | 376 (58.0) |
| Unemployed | 272 (42.0) |
| **Primary house** |  |
| Current house | 235 (36.6) |
| Another province/rural | 225 (35.1) |
| Another country | 181 (28.2) |
| **Duration at current house** |  |
| Less than 1 year | 123 (19.0) |
| 1-5 years | 212 (32.7) |
| More than 5 years | 314 (48.4) |
| **Number of child dependants** |  |
| None | 381 (58.4) |
| 1 child | 121 (18.6) |
| 2 or more children | 150 (23.0) |
| **Number of sexual partners in the past 12 months** |  |
| None | 64 (9.9) |
| 1 Partner | 378 (58.8) |
| >=2 partners | 201 (31.3) |
| **HIV testing history** |  |
| <=12 months ago | 203 (31.6) |
| >12 months ago | 287 (44.6) |
| Never tested for HIV before current test | 153 (23.8) |
| **Person to whom intention to test was disclosed** |  |
| Partner/spouse | 224 (34.8) |
| Family/Friends/Other | 177 (27.5) |
| No one | 242 (37.6) |
| **Support at clinic for latest HIV test** |  |
| Partner/spouse | 76 (11.8) |
| Family/other | 91 (14.2) |
| No one | 476 (74.0) |

*(Continued)*

**Table 1.** (Continued)

| | Total |
|---|---|
| | **N = 652 (col%)** |
| **Intention to disclose HIV test result** | |
| Yes | 618 (96.3) |
| No | 24 (3.7) |
| **Perceived social support** | |
| Medium to high | 612 (95.3) |
| Low | 30 (4.7) |
| **Concerns regarding ART** | |
| Low | 307 (47.7) |
| Medium | 335 (52.1) |
| High | 1 (0.2) |
| **Intention to start ART** | |
| Yes | 631 (99.4) |
| No | 4 (0.6) |
| **Initiated on ART up to 6 months post-test** | |
| No | 126 (19.3) |
| Yes | 526 (80.7) |

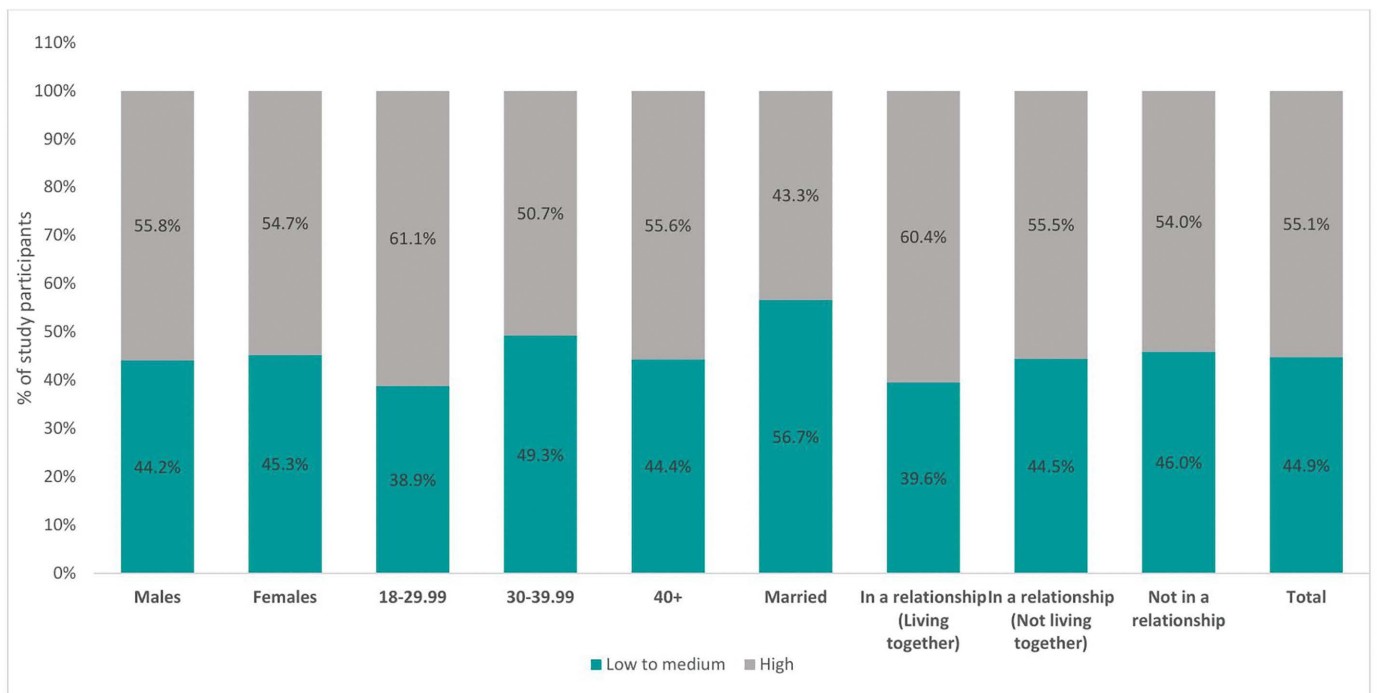

**Fig 1. Prevalence of high anticipated HIV stigma.**

in the same population. Regarding age, the majority (50.7%) of participants with high anticipated stigma were aged between 30 and 40 years, 35% were aged between 18 and 29 years, and 24% were older than 40 years at the time of enrollment. Regarding marital status, 43% of those who were married, 60% of those who were in a cohabiting relationship, 56% of those who were in a non-cohabiting relationship, and 54% of those who were single had high anticipated stigma.

Furthermore, we present median scores by gender, age, and marital status. The median score for anticipated HIV stigma was similar for both males and females (median = 2.6, IQR: 2.2–3) The distribution of scores was similar across the different age groups (median = 2.6, IQR: 2.2–3) (**Fig 2**). However, when looking at the marital status, the median score for anticipated stigma was lower among married participants (median = 2.4, IQR: 2.2–3) compared to the median of 2.6 for those who were in relationships (**Fig 3**).

### Correlates of high anticipated HIV stigma among newly diagnosed

After adjusting for demographic patients' characteristics, unmarried individuals who were in a relationship were more likely to have high anticipated stigma than married participants (aRR 1.10, 95% CI: 1.01-1.18). High anticipated stigma was lower among: older individuals (aRR 0.94 for being 30–39 vs 18–29 years, 95% CI: 0.88-0.99), those having a primary house in another province/rural (aRR 0.82 for primary house in another country vs current house, 95% CI: 0.78-0.87), (aRR 0.83 for primary house in another country vs current house, 95% CI: 0.78-0.88), those living in current homes for ≥5 years (aRR 0.93 for >5 years vs < 1 year, 95% CI: 0.88-0.99), those with low ART concerns (aRR 0.86, 95% CI: 0.82-0.90), and those with low perceived social-support (aRR 0.79 for low vs high, 95% CI: 0.70-0.88). However, gender, level of education, person to whom they had they had intended to disclose, reason for HIV testing, and level of depression did not show association with anticipated stigma among the newly diagnosed Table 2.

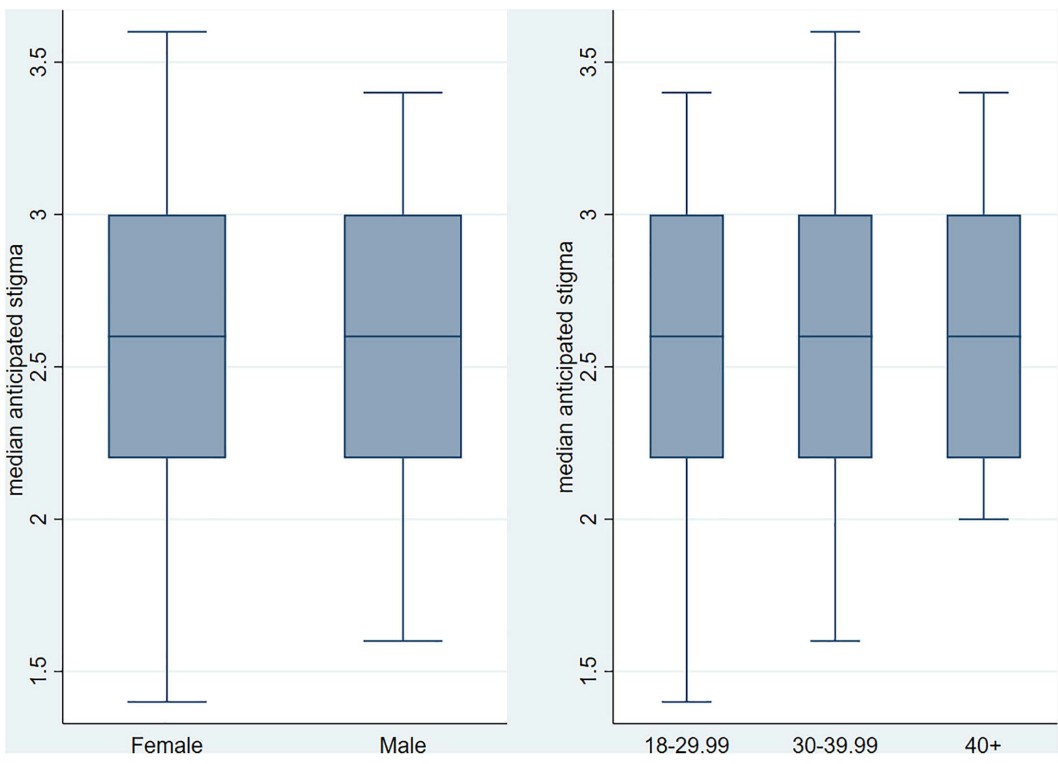

**Fig 2. Median anticipated HIV stigma by sex and age group.**

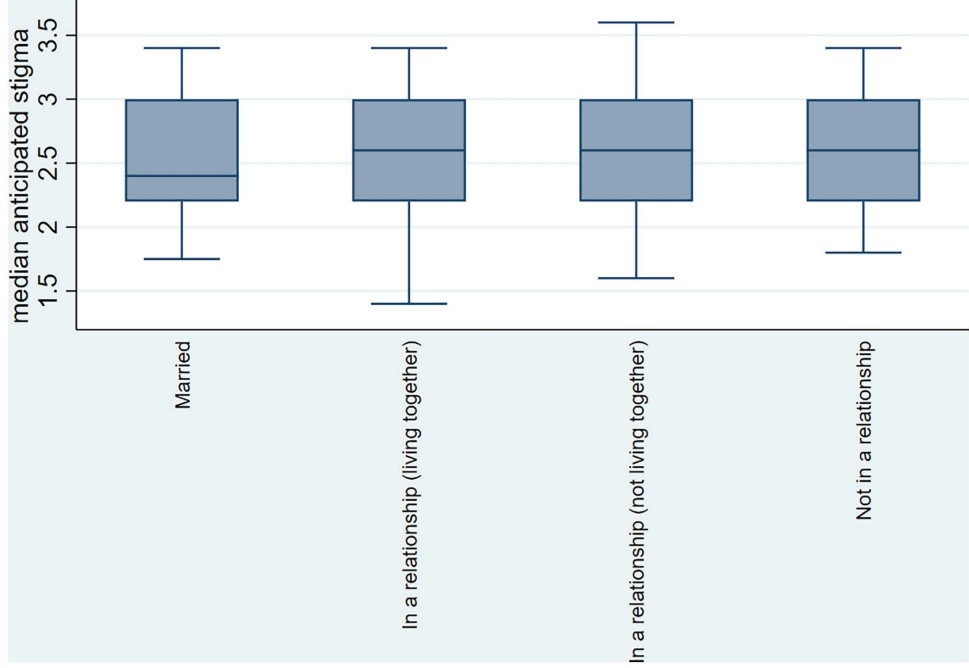

**Fig 3. Median anticipated HIV stigma by marital status.**

## Discussion

The current study measured the prevalence of anticipated stigma among patients newly diagnosed with HIV between October 2017 to August 2018. Our results indicate that over 50% of this population had high anticipated stigma, which is a cause for concern. This result indicates that despite the scale of treatment in South Africa and advances to make HIV more manageable, the expectation of social stigma remains a challenge [34].

The findings also showed similarly high anticipated stigma for male and female participants. These findings are consistent with findings from studies in South Africa, which indicated that both males and females had high levels of external stigma [35,36]. In other settings, however, several studies have shown stigma to be higher in women compared to men [37–39], as women are traditionally held to higher moral standards than men and expected to uphold the moral foundation of society [40]. But other studies did not find gender differences in relation to anticipated stigma [41]. This could be explained by studies that recruit participants from the clinic-based settings where men and women might already be accessing HIV care. These shared experiences may reduce perceived stigma differences by gender. Furthermore, in settings where treatment access and awareness are high, stigma may be declining in both genders, making it harder to detect if there are any differences.

Furthermore, unmarried individuals who were in a relationship reported higher anticipated stigma than those who were married. Previous research has shown that anticipated stigma is significantly associated with non-disclosure of HIV status, particularly among women. This barrier is especially pronounced among women who are not in stable relationships, where fear of rejection or negative consequences can discourage disclosure [42,43]. While stigmatizing attitudes held by their communities may affect all living with HIV, those who are married or living with a stable partner may have additional emotional and social support that acts as a barrier to community-level prejudice [44,45]. However, other studies report that married persons also face anticipated stigma. Their concerns or anxieties include the fear of stigmatisation, the fear of divorce or rejection by their partners, intimate partner violence and the fear that they may be accused of infidelity [46].

**Table 2. Correlates of high anticipated HIV stigma among newly diagnosed.**

| | High anticipated HIV stigma | RR | aRR |
|---|---|---|---|
| | No. (%) | (95% CI) | (95% CI) |
| **Sex** | | | |
| Female | 225 (54.7) | 1 | |
| Male | 129 (55.8) | 1.01 (0.96-1.06) | |
| **Age at HIV diagnosis, years** | | | |
| 18-29.99 | 124 (61.1) | 1 | 1 |
| 30-39.99 | 145 (50.7) | 0.94 (0.88-0.99) | 0.94 (0.88-0.99) |
| 40+ | 85 (55.6) | 0.97 (0.90-1.03) | 1.00 (0.94-1.07) |
| **Marital status** | | | |
| Married | 39 (43.3) | 1 | 1 |
| In a relationship (Living together) | 136 (60.4) | 1.12 (1.03-1.21) | 1.10 (1.01-1.18) |
| In a relationship (Not living together) | 111 (55.5) | 1.08 (1.00-1.18) | 1.09 (1.00-1.18) |
| Not in a relationship | 68 (54.0) | 1.07 (0.98-1.18) | 1.05 (0.96-1.15) |
| **Highest education level** | | | |
| <Grade 12 | 251 (53.7) | 1 | |
| >= Grade 12 | 103 (58.9) | 1.03 (0.98-1.09) | |
| **English literacy** | | | |
| I can read very well | 201 (57.6) | 1 | |
| I can read somewhat | 121 (53.5) | 0.97 (0.92-1.03) | |
| I cannot read | 32 (49.2) | 0.95 (0.87-1.03) | |
| **Employment status** | | | |
| Employed | 195 (52.4) | 1 | |
| Unemployed | 159 (58.9) | 1.04 (0.99-1.10) | |
| **Primary source of income/finances** | | | |
| Paid job, salary or business | 213 (53.1) | 1 | |
| Spouse/ partner | 68 (61.3) | 1.05 (0.99-1.12) | |
| Parents/ relatives/ friends/other | 72 (56.3) | 1.02 (0.96-1.09) | |
| **Primary House** | | | |
| Current house | 175 (75.1) | 1 | 1 |
| Another province/rural | 97 (43.5) | 0.82 (0.78-0.87) | 0.82 (0.78-0.87) |
| Another country | 76 (42.7) | 0.81 (0.77-0.87) | 0.83 (0.78-0.88) |
| **Duration living in current suburb/town or community** | | | |
| Less than 1 year | 74 (61.7) | 1 | 1 |
| 1-5 years | 122 (57.8) | 0.98 (0.91-1.05) | 0.98 (0.92-1.04) |
| More than 5 years | 158 (50.8) | 0.93 (0.87-0.99) | 0.93 (0.88-0.99) |
| **Breadwinner of household** | | | |
| Yes | 178 (54.4) | 1 | |
| No | 174 (55.6) | 1.00 (0.96-1.06) | |
| **Access to basic necessities (amenities score)** | | | |
| Low | 17 (47.2) | 1 | |
| Medium | 134 (53.8) | 0.94 (0.83-1.05) | |
| High | 194 (57.2) | 0.98 (0.93-1.03) | |
| **Lives with** | | | |
| Partner/spouse | 159 (56.2) | 1 | |
| Family/friends | 82 (55.8) | 1.00 (0.94-1.06) | |
| Alone | 76 (51.0) | 0.97 (0.91-1.03) | |

*(Continued)*

| | High anticipated HIV stigma | RR | aRR |
|---|---|---|---|
| | No. (%) | (95% CI) | (95% CI) |
| **Number of child dependants** | | | |
| None | 198 (52.7) | 1 | 1 |
| 1 child | 75 (62.5) | 1.06 (1.00-1.13) | 1.03 (0.97-1.09) |
| 2 or more children | 81 (55.5) | 1.02 (0.96-1.08) | 0.99 (0.93-1.05) |
| **Recent clinic attendance (any)** | | | |
| Never | 66 (61.7) | 1 | |
| within a year | 174 (51.3) | 0.94 (0.88-1.00) | |
| More than a year ago | 114 (58.2) | 0.98 (0.91-1.05) | |
| **HIV and ART knowledge** | | | |
| Low to medium | 51 (49.0) | 1 | |
| high | 303 (56.3) | 1.05 (0.98-1.12) | |
| **Number of sexual partners in the past 12 months** | | | |
| None | 30 (46.9) | 1 | 1 |
| 1 Partner | 194 (51.5) | 1.03 (0.94-1.12) | 1.02 (0.93-1.12) |
| >=2 partners | 128 (64.3) | 1.12 (1.02-1.23) | 1.06 (0.96-1.17) |
| **Condom use at last sex** | | | |
| Yes | 120 (57.4) | 1 | |
| No | 234 (54.0) | 0.98 (0.93-1.03) | |
| **Last HIV test before current test** | | | |
| last HIV test <=12 months ago | 101 (50.0) | 1 | 1 |
| last HIV test >12 months ago | 170 (59.4) | 1.06 (1.00-1.13) | 1.05 (0.99-1.11) |
| Never tested for HIV before | 82 (53.6) | 1.02 (0.96-1.10) | 1.02 (0.95-1.08) |
| **Person to whom they disclosed intention to test for HIV** | | | |
| partner/spouse | 133 (59.4) | 1 | |
| Family/Firends/Other | 88 (49.7) | 0.94 (0.88-1.00) | |
| No one | 133 (55.2) | 0.97 (0.92-1.03) | |
| **Person accompanying to the clinic for current HIV test** | | | |
| Partner/spouse | 51 (67.1) | 1 | 1 |
| Family/other | 50 (54.9) | 0.93 (0.85-1.12) | 0.92 (0.84-1.00) |
| No one | 253 (53.3) | 0.92 (0.86-0.98) | 0.95 (0.89-1.02) |
| **Intention to disclose** | | | |
| Yes | 339 (54.9) | 1 | |
| No | 15 (62.5) | 1.05 (0.93-1.19) | |
| **ART concerns** | | | |
| Medium ARV concerns | 220 (67.7) | 1 | 1 |
| Low ARV concerns | 134 (42.3) | 0.85 (0.81-0.89) | 0.86 (0.82-0.90) |
| **Reasons for HIV testing** | | | |
| Just to know HIV status | 70 (62.5) | 1 | |
| Current/previous HIV risk | 55 (57.3) | 0.97 (0.89-1.05) | |
| Symptomatic | 216 (53.5) | 0.94 (0.89-1.01) | |
| **Depression** | | | |
| No depression | 315 (54.6) | 1 | |
| Major depression | 36 (63.2) | 1.10 (0.97-1.14) | |

*(Continued)*

**Table 2.** (Continued)

| | High anticipated HIV stigma | RR | aRR |
|---|---|---|---|
| | No. (%) | (95% CI) | (95% CI) |
| **Perceived social support** | | | |
| Medium to high | 348 (56.9) | 1 | 1 |
| Low | 6 (20.0) | 0.77 (0.68-0.86) | 0.79 (0.70-0.88) |
| **Expectation of social support after HIV disclosure** | | | |
| Low | 14 (63.6) | 1 | |
| Medium to high | 340 (54.8) | 0.95 (0.83-1.07) | |
| **Baseline CD4 count at testing** | | | |
| <350 | 99 (52.7) | 0.97 (0.89-1.06) | |
| 350-500 | 36 (63.2) | 1.04 (0.94-1.16) | |
| >500 | 41 (56.9) | 1 | |
| Missing | 178 (54.8) | 0.99 (0.91-1.07) | |
| **Initiated on ART** | | | |
| Initiated on ART | 296 (56.9) | 0.94 (0.88-1.00) | |
| No ART initiation | 58 (47.5) | 1.57 (1.53-1.16) | |

Unmarried individuals might worry more about social stigma and be more afraid to disclose because there's a chance they won't find future sexual partners, and those with partners may fear being rejected and losing those relationships [47]

Conversely, older participants (30–39) reported a lower risk of anticipated stigma compared to younger participants, and this finding is consistent with previous findings that normalisation of HIV as a chronic condition, whereby older people compare HIV to other chronic conditions such as hypertension and diabetes [48]. Furthermore, young people are more likely to get into more relationships and have more potential sexual partners, and may experience more fears of disclosure and may amplify anticipated stigma due to worries about rejection, and relationships coming to an end upon disclosure [49]. This underscores the importance of targeted messaging around the Undetectable = Untransmittable (U = U), which has shown potential to reduce stigma [50]. Similarly, those with low ART concerns were less likely to report anticipated stigma, this could be explained by the fact that they have higher factual knowledge of HIV/ART, as we have reported in our previous publication [51]. Knowledge was shown to be associated with reduced concerns, as they are more aware of their diagnosis and treatment and generally have a more positive outlook.

Furthermore, those with low perceived social support were also less likely to report anticipated stigma. This could be as a result of them having developed coping mechanisms that help them handle stigma better and the potential for alternative sources of validation. However, this finding should be interpreted with caution. Prior research shows that social support from friends and family can buffer against stigma and improve psychosocial outcomes, whereas family disapproval or conflict can exacerbate stigma and result in isolation. [52,53]. The interaction between social networks and anticipated stigma is complex and depends on the quality rather than the simple presence.

High levels of stigma necessitate comprehensive interventions that address psychosocial and structural determinants of stigma. Evidence underscores the effectiveness of community-based and culturally appropriate interventions, the integration of U = U, and the strengthening of anti-discrimination laws, and actively engaging PLHIV when designing interventions to mitigate stigma.

## Study limitations

Firstly, we only surveyed HIV positive patients who had engaged in healthcare services; it is possible that some HIV positive people are more likely to report anticipated stigma and would have completely avoided testing, and

this could explain the inability to observe some of the previously reported factors. Secondly, we interviewed patients immediately after their post-test counselling and it is possible that some may have, and it is possible that some may not have had sufficient time to adequately process their diagnosis and think about other aspects of their lives. Thirdly, we dis not cluster the Poisson regression by clinic because there are only five different clinics in the sample. When the number of clusters is this small, the cluster-robust standard errors become unstable and can give misleading results.

## Conclusions

Our findings show that over 50% of adults diagnosed with HIV in the UTT era report high anticipated stigma. These findings suggest that stigma remains pervasive and highlight the need to address factors that continue to drive anticipated stigma, to mitigate the potential impact on engagement in HIV care. HIV programs need to be more focused on strategies that directly address stigma upon diagnosis.

## Supporting information

**S1 Text. Study questionnaire on perceptions about disclosure of HIV status.**
(DOCX)

**S1 Checklist. STROBE checklist.**
(DOCX)

## Acknowledgments

We extend our gratitude to the data collection team (Julia Mogwasi, Jan Maobela, Nokukhanya Mhlanga, Vinolia Njinkelanga, Zoleka Luvuno, Nonhlanhla Tshabalala) for their diligent support during the study implementation. Additionally, our sincere thanks go to the City of Johannesburg and the staff of participating clinics for accommodating the study. We also thank patients attending these clinics for their willingness to participate and share their valuable information.

## Author contributions

**Conceptualization:** Idah Mokhele, Dorina Onoya.

**Data curation:** Tembeka Sineke, Idah Mokhele, Dorina Onoya.

**Formal analysis:** Tembeka Sineke, Idah Mokhele.

**Funding acquisition:** Dorina Onoya.

**Investigation:** Tembeka Sineke, Idah Mokhele, Dorina Onoya.

**Methodology:** Tembeka Sineke, Idah Mokhele, Dorina Onoya.

**Project administration:** Tembeka Sineke.

**Supervision:** Dorina Onoya.

**Validation:** Idah Mokhele.

**Writing – original draft:** Tembeka Sineke.

**Writing – review & editing:** Tembeka Sineke, Idah Mokhele, Robert AC Ruiter, Dorina Onoya, Mandisa Dukashe.

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
