## [Decision Letter · Decision Letter 0]

21 Aug 2025

PGPH-D-25-01312

Impact of UTT on anticipated stigma among patients newly diagnosed with HIV in Johannesburg, South Africa

Dear Dr. Onoya,

Thank you for submitting your manuscript to PLOS Global Public Health. After careful consideration, we feel that it has merit but does not fully meet PLOS Global Public Health’s publication criteria as it currently stands. Therefore, we invite you to submit a revised version of the manuscript that addresses the points raised during the review process.

We look forward to receiving your revised manuscript.

Kind regards,

Guillaume Fontaine, PhD, RN

Academic Editor

Journal Requirements:

1. Please provide additional details regarding participant consent. In the ethics statement in the Methods and online submission information, please ensure that you have specified (1) whether consent was informed and (2) what type you obtained (for instance, written or verbal, and if verbal, how it was documented and witnessed).

https://www.tandfonline.com/doi/full/10.1080/09540121.2021.1902927

In your revision ensure you cite all your sources (including your own works), and quote or rephrase any duplicated text outside the methods section. Further consideration is dependent on these concerns being addressed.

3. In the online submission form, you indicated that [The de-identified data supporting the findings of this study are available from the corresponding author upon reasonable request, subject to ethical approvals and data use agreements in place.].

a. In a public repository,

b. Within the manuscript itself, or

c. Uploaded as supplementary information.

Additional Editor Comments (if provided):

Dear authors,

Thank you for submitting your manuscript to PLOS Global Public Health. In addition to the reviewer comments, please address the following points:

1. Your cover letter describes a longitudinal cohort and impact/trends, but the manuscript and abstract describe a cross-sectional survey. Either (A) truly analyze pre- vs post-UTT with repeated measures/time trends or (B) reframe as prevalence and correlates in the UTT era. Suggested title: Anticipated HIV stigma among newly diagnosed adults in Johannesburg in the UTT era: a cross-sectional study.

2. Methods: STROBE (cross-sectional) is the primary guideline for your study design; complete it, attach it as supplementary material and ensure complete reporting of design, setting, variables, bias, study size, statistical methods, and participant flow.

3. Methods: Name and cite the five items used, show wording (in an appendix), and justify any adaptations.

4. Methods: Justify Modified Poisson with robust SE and state whether you clustered by clinic. It would also be important to pre-specify a confounder set (conceptual framework/DAG) and keeping it the same across models. How did you address missing data?

5. Results: You report lower anticipated stigma among those with low perceived social support—this likely reflects reverse coding or misclassification. Audit coding and table labels (e.g., “low vs high”).

6. Results: Please double-check the “Primary house” categories and percentages (some proportions look implausible relative to the 55% overall).

7. Results: Ensure all variables mentioned (e.g., depression) are fully described in Methods (instrument, scoring, cut-points).

Thank you,

Academic Editor Guillaume Fontaine

Reviewers' comments:

Reviewer's Responses to Questions

**Comments to the Author**

1. Does this manuscript meet PLOS Global Public Health’s publication criteria?

Reviewer #1: Yes

2. Has the statistical analysis been performed appropriately and rigorously?

Reviewer #1: Yes

3. Have the authors made all data underlying the findings in their manuscript fully available (please refer to the Data Availability Statement at the start of the manuscript PDF file)?

Reviewer #1: No

4. Is the manuscript presented in an intelligible fashion and written in standard English?

Reviewer #1: Yes

Reviewer #1: Overall, a well-written study investigating levels of anticipated stigma in people newly diagnosed with HIV. The intro could be developed to elaborate on the novelty of the study. I have made some suggestions below around the interpretation of the findings and the development of the discussion

Abstract

Clarify in the methods that "low-to-medium” or “high” refers to anticipated stigma

It could be more intuitive to report high anticipated stigma was more associated with XYZ

Cronbach’s alpha reported in Abstract but not paper

Introduction

Lines 89-92 “The stigma index…” reference 8 does not correspond. Support the statement with a relevant and current reference. Expand on what internalised stigma means.

Line 103 – clarify if it refers to poor adherence to ART

Line 115 – explain the changes to treatment guidelines

Line 116 – UTT has not been previously spelt out/mentioned/ explained

Introduction needs a further paragraph explaining the need for this study. I think the novelty lies in providing updated data in response to the National Strategic Plan and could provide more information on the South African context of UTT etc. Could cut some of the introcution which is repetitive e.g., Lines 108-111

Methods

Expand on recruitment methods: location, how many sites, what type of sites

Lines 128-130 delete description as it is stated in Results

Line 132 it sounds like a five point scale was used

Results

Not clear why Table 1 is stratified by sex when there is no further stratified analysis. Add justification (in the introduction and/or methods) or just present the overall sample characteristics

Line 159 – says characteristics are presented by gender but the text presents by sex male/female

Line 161 – add time period of recruitment and information on number of recruitment sites

Table 2 presents the same data as Figure 1. Might be better to replace Figure 1 with a figure summarizing the mean score response to each of the five questions. Also provide the five questions in full

Discussion

Line 220 – not clear on the connection between anticipated stigma and receiving counselling

Line 227 – add reference

Lines 234-248 in this section I would just bear in mind that the aRR is just on the cusp of significance, so in the last sentence I don’t think your findings contrast with previous because there is no statistically significant difference between married and single people in this study

Lines 249-256 could explore whether younger people having more anticipated stigma is related to them having more potential partners. Then discuss in relation to raising awareness of U=U

Line 257-261 Could this finding be related to the questions – people aren’t concerned about being rejected/judged because they don’t have friends/family around them? I think there needs to be a more nuanced interpretation here – considering that support from friends/family might counter some types of stigma but disapproval (?) from family can be isolating

Could add a paragraph on policy implications – what can be done to address high prevalence of anticipated stigma

Minor

Spell out universal test and treat in the title

Some typos – capitalize first letter of background paragraph in abstract

Spell out acronyms which are lesser used like “National Strategic Plan”

**Do you want your identity to be public for this peer review?** For information about this choice, including consent withdrawal, please see our Privacy Policy

Reviewer #1: No

---

## [Editor Report · Decision Letter 1]

8 Jan 2026

Impact of Universal Test and Treat (UTT) on anticipated stigma among patients newly diagnosed with HIV in Johannesburg, South Africa: a cross-sectional study

PGPH-D-25-01312R1

Dear Dr. Onoya,

We are pleased to inform you that your manuscript 'Impact of Universal Test and Treat (UTT) on anticipated stigma among patients newly diagnosed with HIV in Johannesburg, South Africa: a cross-sectional study' has been provisionally accepted for publication in PLOS Global Public Health.

Best regards,

Guillaume Fontaine, PhD, RN

Academic Editor